# Advanced Glycation End Products Impair Cardiac Atrial Appendage Stem Cells Properties

**DOI:** 10.3390/jcm10132964

**Published:** 2021-07-01

**Authors:** Lize Evens, Ellen Heeren, Jean-Luc Rummens, Annelies Bronckaers, Marc Hendrikx, Dorien Deluyker, Virginie Bito

**Affiliations:** 1BIOMED, UHasselt—Hasselt University, Agoralaan, 3590 Diepenbeek, Belgium; lize.evens@uhasselt.be (L.E.); ellen.heeren@student.uhasselt.be (E.H.); jeanluc.rummens@uhasselt.be (J.-L.R.); annelies.bronckaers@uhasselt.be (A.B.); marc.hendrikx@uhasselt.be (M.H.); dorien.deluyker@uhasselt.be (D.D.); 2Faculty of Medicine and Life Sciences, UHasselt—Hasselt University, Agoralaan, 3590 Diepenbeek, Belgium

**Keywords:** stem cells, aldehyde dehydrogenase, CASCs, glycated proteins, advanced glycation end products, proliferation, apoptosis, migration, RAGE inhibition

## Abstract

Background: During myocardial infarction (MI), billions of cardiomyocytes are lost. The optimal therapy should effectively replace damaged cardiomyocytes, possibly with stem cells able to engraft and differentiate into adult functional cardiomyocytes. As such, cardiac atrial appendage stem cells (CASCs) are suitable candidates. However, the presence of elevated levels of advanced glycation end products (AGEs) in cardiac regions where CASCs are transplanted may affect their regenerative potential. In this study, we examine whether and how AGEs alter CASCs properties in vitro. Methods and Results: CASCs in culture were exposed to ranging AGEs concentrations (50 µg/mL to 400 µg/mL). CASCs survival, proliferation, and migration capacity were significantly decreased after 72 h of AGEs exposure. Apoptosis significantly increased with rising AGEs concentration. The harmful effects of these AGEs were partially blunted by pre-incubation with a receptor for AGEs (RAGE) inhibitor (25 µM FPS-ZM1), indicating the involvement of RAGE in the observed negative effects. Conclusion: AGEs have a time- and concentration-dependent negative effect on CASCs survival, proliferation, migration, and apoptosis in vitro, partially mediated through RAGE activation. Whether anti-AGEs therapies are an effective treatment in the setting of stem cell therapy after MI warrants further examination.

## 1. Introduction

Coronary heart disease (CHD) remains the leading cause of mortality and morbidity worldwide, with myocardial infarction (MI) being the most common form of CHD [1]. MI results from complete or partial occlusion of a coronary artery. In the ischemic area, oxygen and nutrients are restricted, resulting in myocardial cell death. The infarct size depends on multiple factors, such as the size of the ischemic area at risk, the location and duration of coronary occlusion, and the amount of residual collateral blood flow [1,2]. As adult cardiomyocytes have minimal regenerative properties, intrinsic repair of the damaged tissue remains elusive. Finding a therapeutic approach that effectively replaces myocardial scar with functional contractile tissue is the only option to recover lost cardiac tissue.

Much research effort has been put into unravelling the therapeutic potential and mechanisms of cell therapy with bone marrow cells (BMCs). Bone marrow contains hematopoietic stem cells (HSCs), endothelial progenitor cells (EPCs), and mesenchymal stem cells (MSCs) [3]. Clinical trials with mononuclear BMCs and MSCs failed to deliver significant improvements in post-MI cardiac function. As mononuclear BMCs and MSCs do not differentiate into cardiomyocytes, the limited improvements observed are likely to be attributed to paracrine mechanisms [4,5]. To significantly improve post-MI cardiac function, the focus of research shifted towards resident cardiac stem cells (CSCs) such as c-kit+, Sca-1+-, Isl-1+-cells, and cardiospheres, who are likely to be pre-programmed to become cardiomyocytes. Yet, their success in cardiac regeneration is poor [6].

In the last years, our research group discovered a new type of cardiac stem cells named ‘cardiac atrial stem cells’ (CASCs). In contrast to other stem cells, CASCs display extraordinary cardiomyogenic differentiation properties, making them a promising candidate for cardiac regeneration [4]. Isolation of this stem cell population from atrial appendages is based on high aldehyde dehydrogenase (ALDH) activity. High ALDH activity was also reported in other stem cell types, like MSCs, HSCs, neural and cancer stem cells amongst others [7,8,9]. Since ALDH has proven to be cardioprotective and promotes cell survival in stress conditions, using an ALDH^+^ stem cell population in ischemic conditions may be, in this context, beneficial [4,10]. CASCs can be expanded up to clinically relevant numbers, without losing fundamental characteristics, such as ALDH activity, surface antigen profile, and its cardiomyogenic differentiation capacity [11]. This given is crucial for translation of this therapeutic approach to the clinic. In addition, we have shown that autologous CASCs transplantation results in improved left ventricular function, resulting from adequate stem cell engraftment and further CASCs differentiation [4,12].

In MI patients, levels of advanced glycation end-products (AGEs) are increased [13]. AGEs are proteins and/or lipids that are irreversibly damaged by glycation, a process in which reducing sugars react non-enzymatically with amino groups in lipids or proteins. Besides glycation, oxidative stress also leads to AGEs formation through oxidation of proteins and/or lipids [14]. AGEs are formed endogenously and naturally accumulate in the body with senescence or in pathological situations such as MI, when levels of oxidative stress are increased [15,16,17]. Furthermore, previous research has shown that AGEs affect different types of stem cells in vitro [18,19,20]. The capacity of stem cells to proliferate is reduced by AGEs, and the apoptotic rate is increased upon AGEs application. These effects could be executed throughout several mechanisms, including activation of the apoptotic pathway, RAGE, or excessive ROS formation [20]. Whether AGEs also affect CASCs properties remains unknown. These findings raise the question whether AGEs could negatively influence the therapeutic efficacy of CASCs, which are used as a treatment for MI. The aim of this study is therefore to examine the in vitro effects of AGEs and the potential activation of its receptor RAGE on proliferation, survival, and migration of CASCs.

## 2. Materials and Methods

### 2.1. Animal Experiments

Animal studies were conducted in accordance with the EU Directive 2010/63/EU for animal experiments and approved by the Local Ethical Committee for Animal Experimentation (UHasselt, Belgium, Diepenbeek; ID 201919K). All animals were kept in a temperature-controlled environment (21 °C, 60% humidity) with a 12 h–12 h light-dark cycle. They were fed a standard pellet diet with water available *ad libitum*. In total, 62 female Sprague-Dawley rats (Janvier Labs, Le Genest-Saint-Isle, France) were used.

### 2.2. Rat CASCs Isolation and Expansion

CASCs were harvested from the right atrial appendages, as described before [4]. Briefly, rats were injected with heparin (1000 units/kg, intraperitoneally (i.p.)) and were euthanized with an overdose of sodium pentobarbital (Dolethal, Vetoquinol, Aartselaar, Belgium, 200 mg/kg, i.p.). Hearts were harvested, perfused with a normal Tyrode solution (137 mM NaCl, 5.4 mM KCl, 0.5 mM MgCl_2_, 1 mM CaCl_2_, 11.8 mM Na-HEPES, 10 mM glucose, 20 mM taurine, pH 7.4), and the right atrial appendages were collected. The extracted right atrial appendage tissue was minced in pieces of ~1 mm^3^, washed with phosphate buffered saline (PBS), and enzymatically dissociated for 30 min in Hank’s Balanced Salt Solution containing 0.6 WU/mL collagenase NB 4 (Serva, Heidelberg, Germany) and 20 mM CaCl_2_. ALDH^+^ cells were stained according to the Aldefluor kit (STEMCELL Technologies, Evergem, Belgium). ALDH^+^ cells were defined as CASCs and were flow-sorted (BD FACS Aria) in X-VIVO 15 media (Lonza, Basel, Switzerland) supplemented with 20% fetal calf serum (FCS) and 2% penicillin/streptomycin (P/S). Isolated CASCs were seeded in 6-well plates at a density of 60,000 cells per well and incubated at 37 °C in a humidified incubator with a 5% CO_2_ atmosphere. Medium was changed every 2 to 3 days. When CASCs reached 80% confluence, they were harvested using trypsin. For all experiments, passage 1 CASCs were used.

### 2.3. AGEs Preparation

AGEs were prepared as previously described [21]. Briefly, bovine serum albumin (BSA; 7 mg/mL) was incubated with glycolaldehyde dimers (90 mM; Sigma-Aldrich, Diegem, Belgium) in sterile PBS (pH 7.4) for 5 days at 37 °C. This solution was dialyzed against PBS, two times for 2 h and overnight at 4 °C to remove unreacted glycolaldehyde (3.4 kDa cut-off). AGEs were filtered (0.2 µm filter, Sarstedt, Antwerp, Belgium). BSA incubated in PBS (7 mg/mL) was used as a control solution.

### 2.4. Proliferation and Survival Assay

Proliferation and survival assays were executed, with a propidium iodide (PI) assay as described before by Gervois et al. and Lo Monaco et al. [22,23]. Briefly, CASCs were seeded in a 96-well plate in X-VIVO medium with 10% FCS and 2% P/S. For proliferation assays, 5000 cells per well were seeded. For survival assays, 10,000 cells per well were seeded. After 24 h, five different conditions were added to the medium: 400 µg/mL BSA, 50 µg/mL, 100 µg/mL, 200 µg/mL, and 400 µg/mL AGEs. To measure proliferation, BSA or AGEs were added to X-VIVO medium with 2% FCS and 2% P/S. To measure survival, BSA or AGEs were added to X-VIVO medium with 0% FCS and 2% P/S. After three different time points (24, 48, and 72 h), the medium was replaced with Lysis buffer A100 (ChemoMetec, Kaiserslautern, Germany), followed by an equal amount of stabilization buffer B (ChemoMetec) supplemented with PI (10 µg/mL, Sigma). Following an incubation period of 15 min in the dark, fluorescence was measured using the Fluostar Optima plate reader (BMG Labtech, Ortenberg, Germany) at an excitation of 540 nm, emission wavelength of 612 nm, and a gain of 2000. Experiments were performed in triplicate. Data were normalized to data obtained with 400 µg/mL BSA.

### 2.5. Migration Assay

CASCs were seeded in a 12-well plate at a density of 50,000 cells per well in X-VIVO medium with 10% FCS and 2% P/S. Five conditions were added to the medium: 400 µg/mL BSA, 50 µg/mL, 100 µg/mL, 200 µg/mL, and 400 µg/mL AGEs. After an incubation period of 72 h, the conditioned CASCs were harvested using trypsin and used for a transwell migration assay. In the ThinCerts (Greiner Bio-One, Vilvoorde, Belgium) with a porous membrane of 8 µm pore size, 100,000 cells per condition were seeded in X-VIVO medium with 0% FCS and 2% P/S. ThinCerts were placed onto 24-well plates containing X-VIVO medium with 2% FCS and 2% P/S. After 24 h of migration, the ThinCerts were fixated with 4% paraformaldehyde (PFA) for 15 min and incubated with 0.1% crystal violet for 30 min. Cells that did not migrate were removed at the top side of the ThinCerts, after which the amount of transmigrated CASCs was quantified with AxioVision 4.6 software (Carl Zeiss, Zaventem, Belgium). Data were normalized to data obtained with 400 µg/mL BSA.

### 2.6. Apoptosis Assay

CASCs were seeded in a 96-well plate at a density of 10,000 cells per well in X-VIVO medium with 2% FCS and 2% P/S. To study apoptosis, a caspase assay was performed using the IncuCyte^®^ Caspase-3/7 Green Apoptosis Assay Reagent (diluted 1/1000, Sartorius, Schaarbeek, Belgium). Five conditions were added to the medium: 400 µg/mL BSA, 50 µg/mL, 100 µg/mL, 200 µg/mL, and 400 µg/mL AGEs. CASCs cultured in X-VIVO medium without FCS and 2% P/S were used as a positive control. Experiments were performed in triplicate. Images were taken after 24, 48, and 72 h of incubation using the IncuCyte^®^ S3 Live-Cell Analysis System (Sartorius, Schaarbeek, Belgium). Analysis of the area occupied by apoptotic cells was performed using the IncuCyte^®^ SX1 Live-Cell Analysis System (Sartorius, Schaarbeek, Belgium). Data were normalized to the positive control (+, X-VIVO medium without FCS).

### 2.7. In Vitro RAGE Inhibition

RAGE was inhibited to assess the contribution of RAGE activation in proliferation, survival, and migration of CASCs. Briefly, CASCs were pre-incubated at 37 °C in a 5% CO_2_ incubator with the RAGE antagonist FPS-ZM1 (10 and 25 µM, Calbiochem/Merck, Overijse, Belgium). After 2 h’ pre-incubation, 400 µg/mL AGEs were added. After 24, 48, and 72 h, proliferation and survival were evaluated as described above. After 72 h of incubation, pre-conditioned CASCs were harvested and used for a transwell migration assay as described above. 

### 2.8. Statistics

Statistical analyses were performed using GraphPad Prism 9.0.0 software. Normal distribution of data was assessed with the Shapiro–Wilk test. Normally distributed data were subjected to a one-way ANOVA test with repeated measurements, followed by the Holm–Sidak’s Multiple Comparison test. When data were not normally distributed, the non-parametrical Friedman test was used followed by the Dunn’s Multiple Comparison test. All data are expressed as mean ± standard error of the mean (SEM). A value of *p* < 0.05 was considered statistically significant.

## 3. Results

### 3.1. AGEs Exposure Negatively Affects CASCs Proliferation and Survival

As shown in Figure 1, AGEs significantly and gradually decreased CASCs proliferation over time. The negative impact of AGEs on CASCs proliferation was also concentration-dependent. After 72 h, concentrations of 100 µg/mL, 200 µg/mL, and 400 µg/mL AGEs significantly reduced CASCs proliferation compared to BSA (Figure 1C; 80% ± 7 in 100 µg/mL, 74% ± 3 in 200 µg/mL, and 65% ± 4 in 400 µg/mL AGEs). Application of BSA alone did not affect the proliferative capacity of CASCs (Appendix A).

As shown in Figure 2, rising concentrations of AGEs negatively affected CASCs survival in time. Significant effects of AGEs were observed after 48 (Figure 2B) and 72 h (Figure 2C; 85% ± 3 in 100 µg/mL, 73% ± 3 in 200 µg/mL, and 64% ± 4 in 400 µg/mL AGEs). Application of BSA alone did not affect the survival capacity of CASCs (Appendix A).

### 3.2. Increased AGEs Concentrations Increase CASCs Apoptosis

To elucidate the effect of different AGEs concentrations (50, 100, 200, and 400 µg/mL) on CASCs apoptosis, a caspase assay was performed. The percentage of cells expressing caspase 3/7 was measured at different time points: 24 (Figure 3A), 48 (Figure 3B), and 72 (Figure 3C) h. Apoptotic rate gradually increased over time with increased AGEs concentrations (Figure 3C, 72 h; 77% ± 17 in 400 µg/mL AGEs vs. 18% ± 3 in BSA).

### 3.3. AGEs Exposure Decreases CASCs Migration Capacity

CASCs migration capacity was evaluated with a transwell migration assay after 72 h of incubation with different concentrations of AGEs. In Figure 4A–E, representative examples of CASCs migration after incubation with BSA and different AGEs concentrations (50, 100, 200, and 400 µg/mL) are presented. Quantification of migration is shown in Figure 4F. Compared to BSA, a significant reduction in migration was observed when CASCs were incubated with 400 µg/mL AGEs (Figure 4E,F; 75% ± 5 in 400 µg/mL AGEs).

### 3.4. Deleterious Effects of AGEs in CASCs Are Mediated by RAGE Activation

To evaluate the contribution of RAGE activation in the observed deleterious effects of AGEs, proliferation, survival, and migration of CASCs were assessed after the incubation with the RAGE antagonist FPS-ZM1. Before exposure to 400 µg/mL AGEs, CASCs were pre-incubated for 2 h with 10 or 25 µM FPS-ZM1. Application of FPS-ZM1 alone (10 and 25 µM) did not affect the proliferative capacity nor the survival of CASCs (Appendix A).

CASCs proliferation (Figure 5A–C) was evaluated after 24 (A), 48 (B), and 72 (C) h. As shown in Figure 1, the negative impact on CASCs proliferation by AGEs is significantly blunted after pre-incubation with 25 µM FPS-ZM1 (Figure 5A–C). Indeed, after 24 and 48 h, CASCs proliferation was significantly improved, with 400 µg/mL AGEs exposure (24 h, Figure 5A; 104% ± 8 in 25 µM FPS-ZM1 vs. 79% ± 5 in 400 µg/mL AGEs; 48 h, Figure 5B; 95% ± 5 in 25 µM FPS-ZM1 vs. 70% ± 4 in 400 µg/mL AGEs). After 72 h, the same trend was observed. Proliferation tended to improve when RAGE was inhibited (Figure 5C; 80% ± 8 in 25 µM FPS-ZM1 vs. 67% ± 5 in 400 µg/mL AGEs, *p* = 0.06).

Survival (Figure 6A–C) was evaluated after 24 (A), 48 (B), and 72 (C) h. The negative impact on CASCs survival by AGEs (Figure 2) is significantly improved after pre-incubation with 25 µM FPS-ZM1 (Figure 6A–C). After 24 h, survival significantly improved (Figure 6A; 104% ± 7 in 25 µM FPS-ZM1 vs. 91% ± 4 in 400 µg/mL AGEs). This trend was also observed after 48 h (Figure 6B; 91% ± 5 in 25 µM FPS-ZM1 vs. 81% ± 5 in 400 µg/mL AGEs, *p* = 0.07). 

Representative examples of CASCs migration after 72 h incubation with 400 µg/mL AGEs and FPS-ZM1 are presented in Figure 7A and quantified in Figure 7B. CASCs pre-incubation with 25 µM FPS-ZM1 could prevent the decreased CASCs migration capacity observed with AGEs exposure (Figure 7B; 98% ± 6 in 25 µM FPS-ZM1 vs. 77% ± 8 in 400 µg/mL AGEs, *p* = 0.07).

## 4. Discussion

Our study is the first to show that AGEs affect CASCs properties, namely survival, proliferation, migration, and apoptosis in vitro. Our data demonstrate that these effects are partially mediated through RAGE activation in a dose-dependent manner. 

### 4.1. The Role of AGEs in MI 

Circulatory AGEs levels are significantly elevated in patients with acute MI [24,25]. However, how they are involved in the pathophysiology of MI remains unclear. Reactive oxygen species (ROS) are the main contributors involved in the synthesis of AGEs. Oxidative stress can induce formation of reactive carbonyl compounds and glycoxidation of Amadori products in the Maillard reaction. As such, AGEs are irreversibly formed and accumulated in the heart after MI and are thought to potentially further worsen the adverse cardiac phenotype [26,27]. In addition, neutrophils and activated macrophages, involved in the inflammatory process in MI, are major contributors of AGEs synthesis [28,29]. These immune cells secrete AGEs and are reported as key inducers of AGEs formation in MI. 

### 4.2. Physiological Relevance of AGEs Concentrations

In our study, we tested a broad range of AGEs concentrations (50 to 400 µg/mL). The AGEs concentration used in other in vitro studies investigating the effect of AGEs on stem cells, ranges from 15 to 500 µg/mL [20]. There is a significant variability of used concentrations, but higher AGEs levels generally reflect the physiological plasma levels found in patients suffering from multiple diseases. Indeed, AGEs-albumin concentration in diabetic patients has been shown to range from 50 up to 400 µg/mL [30]. AGEs levels can rise to concentrations up to 200 µg/mL in patients suffering from cardiovascular diseases [31,32]. In patients with early-stage Alzheimer’s disease, lower AGEs concentrations in the nanoscale range, are also reported [33]. However, because of the different analytical methods used for measuring AGEs, and the heterogeneity of different types of AGEs, estimation of reliable AGEs concentrations in vivo remains technically challenging and is probably an underestimation [34]. 

### 4.3. AGEs Have a Negative Impact on CASCs Properties 

Even if CASCs transplantation shows promising potential for cardiac regeneration post-MI, survival and regenerative capacity of cells remains an issue. Ischemic areas are known to be a hostile environment with increased levels of oxidative stress, inflammation, and fibrosis combined with increased AGEs tissue levels. Whether AGEs would affect the regenerative capacity of CASCs was unknown but could be important knowledge in the context of cardiac regeneration and the promising regenerative capacities of CASCs [12]. In our study, we show that AGEs impair CASCs survival, proliferation, and migration in vitro, in a concentration- and time-dependent manner. Furthermore, exposure to AGEs leads to a gradual increase in CASCs apoptosis. Our data are in line with studies examining the effect of AGEs on multiple types of other stem cells, in which proliferative capacity is altered and apoptosis is increased [20]. Indeed, Zhu et al. demonstrated a significant decrease in EPCs proliferation after exposure to different concentrations of AGEs [16]. The same effect was evaluated by Sun et al. also in EPCs, where an increase in apoptotic rate was mediated by p38 MAPK pathway activation [35]. NSCs exposed to AGEs resulted in a dose-dependent reduction of stem cell proliferation, mediated via the PPARγ pathway [36]. In adipose tissue-derived stem cells (ADSCs), an increase in caspase 3 activation leads to an increased apoptotic rate [37]. Yang et al. reported lower proliferation and migration capacities in MSCs, in an AGEs concentration-dependent way. This effect was mediated via excessive ROS production [18]. Whether the deleterious effects on CASCs, a very different stem cell population of cardiac origin, are also mediated through excessive ROS production, remains to be determined.

It has been shown that the underlying mechanisms in which AGEs execute their negative effects on organ function are dependent and/or independent of RAGE receptor activation [15]. Studies in many stem cells types show that AGEs mediate their effects mainly through the activation of RAGE or other apoptotic pathways [20]. RAGE activation by AGEs causes activation of MAPK, which leads to phosphorylation of JNK and p38 [38]. These phosphorylated proteins increase the transcription of different pro-apoptotic transcription factors in the nucleus, leading to an increase in apoptosis. Next to that, caspase pathways can be activated, causing AGEs-induced apoptosis [39]. Follow-up studies remain necessary to unravel the molecular mechanisms by which the downstream effects of AGEs are induced in CASCs. However, we have shown that upon RAGE-blocking by FPS-ZM1, the observed effects of AGEs on CASCs were blunted. Therefore, our data strongly indicate that AGEs mediate their effects on CASCs likely through binding and activation of RAGE. Whether the Jak/STAT, PI3K/Akt, MAPK, excessive ROS production, or other signaling pathways are involved, remains to be further identified. Our data are also in line with the work described by Zhang et al., where FPS-ZM1 also reversed the negative effects of AGEs in ADSCs by blocking RAGE, further confirming the important role of RAGE activation as a mediator of the deleterious effects caused by AGEs [40].

### 4.4. Future Perspectives for Anti-AGEs Therapies for Cardiovascular Diseases and Current Limitations

In vivo confirmation on the use of anti-AGEs therapies and their potential added value to stem cell transplantation after MI need to be confirmed in an animal model before this can be translated to the clinic. It has been shown in multiple in vitro studies that the PPARγ inhibitor rosiglitazone [41,42], MAPK inhibitors [18,35,43], or antioxidants [44] can attenuate the AGEs-mediated effects on stem cells. However, their influence in vivo as a therapeutic intervention in combination with stem cell transplantation has never been addressed so far. There are multiple strategies to lower AGEs levels in the body. Pyridoxamine (PM) is an inhibitor of AGEs formation by decreasing the Amadori-to-AGEs conversion and scavenging carbonyl compounds. The efficacy as well as the safety of PM treatment has been demonstrated in clinical trials with diabetic patients [45]. However, a clinical trial of NephroGenex in 2014, testing Pyridorin^®^ (i.e., PM) as an anti-diabetic therapy, was stopped due to financial issues [46]. No other clinical trials are currently investigating PM as a therapy. However, inhibiting AGEs formation with PM could be a strategy to improve stem cell potential for cardiac regenerative purposes. In addition, other inhibitors of AGEs formation, like aminoguanidine, could be used in the future to lower AGEs levels after MI. The ACTION II clinical trial showed the efficacy of aminoguanidine in diabetic patients. While aminoguanidine failed to significantly reduce the primary endpoint of doubling the time to reach maximum serum creatinine levels in these patients, other clinically important effects on the complications of diabetes, such as reduction in proteinuria and circulatory lipid concentrations, were shown. However, due to reversible adverse effects such as the induction of autoantibodies, flu-like symptoms, and anemia, this trial was terminated and translation into the clinic remains limited [47,48]. In addition, using antioxidants such as N-acetyl-L-cysteine (NAC) or glutathione as a supplement to our diet could provide some beneficial outcome for stem cell therapy, as they increase genomic stability, improve adhesion, and stimulate stem cell proliferation [49]. However, cell-specific actions differ between stem cell types, and dose-response clinical trials are needed to evaluate their therapeutic efficacy when used in combination with stem cell transplantation. Another option is to break down AGEs with ALT-711 therapy. ALT-711 is able to cleave carbon-carbon bounds between carbonyls, thereby breaking cross-links in AGEs molecules. However, several clinical trials could not confirm the beneficial effects of ALT-711 observed in animal studies. Furthermore, inhibitors of RAGE (like FPS-ZM1) or inhibitors of downstream molecules in the RAGE pathway can interfere in the AGEs/RAGE cell signaling axis, thereby blocking AGEs-mediated effects in stem cells. The efficacy of different types of small molecules and inhibitors to block AGEs in stem cells has been demonstrated in multiple in vitro experiments [18,35,41,42,43,44], but has never been tested before in animal models. Therefore, we can only hypothesize that these inhibitors are efficient in blocking AGEs in an in vivo situation, but proof-of-concept experiments are needed. Finally, another option in blocking AGEs in combination with stem cell therapy, is genetically modifying stem cells itself. Overexpression of sRAGE is known to enhance AGEs scavenging (and other RAGE ligands like amyloid-β) to improve the effectiveness of cell therapy. This has been shown in sRAGE-secreting MSCs as a therapy for Alzheimer’s disease [50,51], arthritis [52], and Parkinson’s disease [53]. sRAGE secreting MSCs survived longer, had enhanced migration capacity, were better protected against apoptosis, and had anti-inflammatory properties. Also, downregulation of RAGE, thereby desensitizing the stem cells for AGEs, could be an option in improving functionality of the cells. Whether these strategies could also be applicable in the setting of MI and cardiac repair remains to be investigated. To summarize, all these strategies aim at tackling AGEs to improve stem cell functionality and retention. However, these therapeutic options remain hypothetical and need to be investigated in vivo in combination with CASCs therapy before this can be translated into the clinical setting. 

## 5. Conclusions

We found that AGEs had a time- and concentration-dependent, gradual effect on CASCs’ properties, by increasing apoptosis and by reducing survival, proliferation, and migration in vitro. The working mechanisms behind these effects remain to be further investigated, although we have shown that RAGE activation is an important contributor of these AGEs-related negative effects. Whether targeting AGEs in vivo could improve CASCs’ therapeutic capacity after MI, remains to be further investigated.

## Figures and Tables

**Figure 1 jcm-10-02964-f001:**
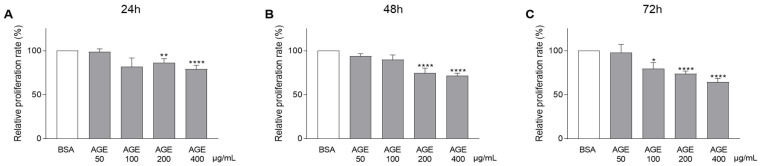
Rising concentrations of AGEs gradually reduce CASCs proliferation. CASCs were exposed to different AGEs concentrations (50, 100, 200, and 400 µg/mL). Proliferation was measured at different time points: 24 h ((**A**), *n* = 18), 48 h ((**B**), *n* = 18), and 72 h ((**C**), *n* = 21). Data are normalized to BSA and represented as mean ± SEM. * denotes *p* < 0.05, ** denotes *p* < 0.01, **** denotes *p* < 0.0001 vs. BSA.

**Figure 2 jcm-10-02964-f002:**
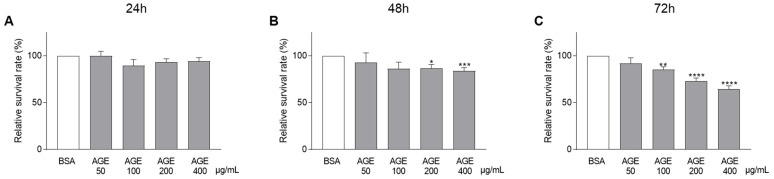
Rising concentrations of AGEs reduced CASCs survival over time. CASCs were exposed to different AGEs concentrations (50, 100, 200, and 400 µg/mL). Survival was measured at different time points: 24 h ((**A**), *n* = 28), 48 h ((**B**), *n* = 20), and 72 h ((**C**), *n* = 25). Data are normalized to BSA and represented as mean ± SEM. * denotes *p* < 0.05, ** denotes *p* < 0.01, *** denotes *p* < 0.001, **** denotes *p* < 0.0001 vs. BSA.

**Figure 3 jcm-10-02964-f003:**
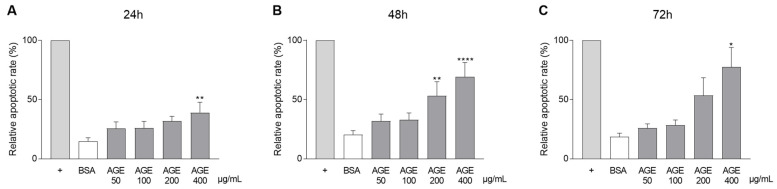
Rising AGEs concentrations increases CASCs apoptosis. CASCs were conditioned with BSA and rising concentrations of AGEs (50, 100, 200, and 400 µg/mL). The percentage of apoptosis was measured over time for 24 ((**A**), *n* = 6), 48 ((**B**), *n* = 8), and 72 h ((**C**), *n* = 7). Data are normalized to the positive control (+, X-VIVO 0% FCS, 2% P/S) and shown as mean ± SEM. * denotes *p* < 0.05, ** denotes *p* < 0.01, **** denotes *p* < 0.0001 vs. BSA.

**Figure 4 jcm-10-02964-f004:**
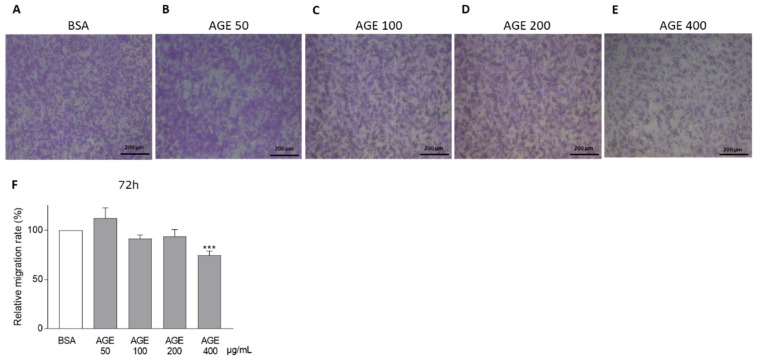
AGEs reduce CASCs migration capacity. Representative examples of CASCs migration when incubated for 72 h with (**A**) BSA and different AGEs concentrations: (**B**) 50 µg/mL, (**C**) 100 µg/mL, (**D**) 200 µg/mL, and (**E**) 400 µg/mL. Scale bar = 200 µm. (**F**) Quantification of CASCs migration after 72 h of incubation with different AGEs concentrations. Fifty µg/mL (*n* = 6), 100 µg/mL (*n* = 10), 200 µg/mL (*n* = 6), and 400 µg/mL AGEs (*n* = 20). Data are normalized to BSA and shown as mean ± SEM. *** denotes *p* < 0.001 vs. BSA.

**Figure 5 jcm-10-02964-f005:**
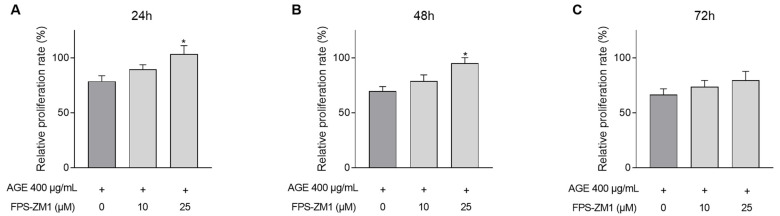
CASCs proliferation is improved when blocking RAGE. CASCs were exposed to 400 µg/mL AGEs. In addition, CASCs were pre-incubated with 10 or 25 µM FPS-ZM1. Proliferation was measured at different time points: 24 ((**A**), *n* = 13), 48 ((**B**), *n* = 12), and 72 h ((**C**), *n* = 15). Data are normalized for BSA and represented as mean ± SEM. * denotes *p* < 0.05 vs. 400 µg/mL AGEs.

**Figure 6 jcm-10-02964-f006:**
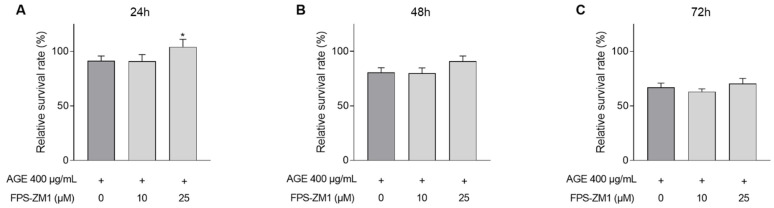
RAGE inhibition improved CASCs survival. CASCs were exposed to 400 µg/mL AGEs. In addition, CASCs were pre-incubated with 10 or 25 µM FPS-ZM1. Survival was measured at different time points: 24 ((**A**), *n* = 14), 48 ((**B**), *n* = 12), and 72 h ((**C**), *n* = 15). Data are normalized for BSA and represented as mean ± SEM. * denotes *p* < 0.05 vs. 400 µg/mL AGEs.

**Figure 7 jcm-10-02964-f007:**
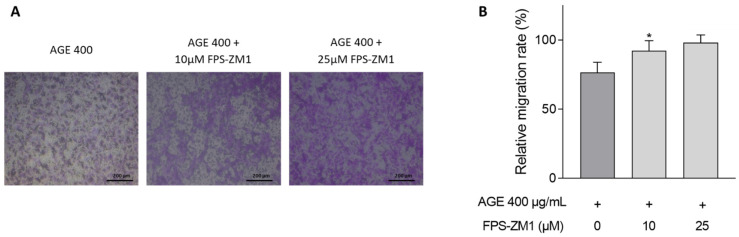
RAGE inhibition improves CASCs migration. (**A**) Representative examples of CASCs migration when incubated for 72 h with 400 µg/mL AGEs and pre-incubated with 10 or 25 µM FPS-ZM1. Scale bar = 200 µm. (**B**) Quantification of CASCs migration (*n* = 11) when incubated with 400 µg/mL AGEs and pre-incubation with 10 or 25 µM FPS-ZM1 for 72 h. Data are normalized for BSA and represented as mean ± SEM. * denotes *p* < 0.05 vs. 400 µg/mL AGEs.

## Data Availability

The data presented in this study are available on request from the corresponding author.

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
