# Peer review of "Advanced Glycation End Products Impair Cardiac Atrial Appendage Stem Cells Properties"

_jcm, 2021, doi:10.3390/jcm10132964_

Round 1

Reviewer 1 Report

The manuscript by Evens et al. entitled Advanced Glycation End Products Impair Cardiac Atrial Appendage Stem Cells Properties” has an interesting concept. However, several aspects need to be further addressed.

1. Is there a reason why only female rats were used for this study? Are there any sex-related differences that are important for the harvesting CASCs?

2. Regarding the proliferation assay in Figure 1, the untreated controls (BSA) are set at 100%. Does that mean that 100% of CASCs is proliferating at the baseline? How exactly was the quantification done?

 3. In Figure 3, what represents the first bar indicated by “+”? This needs to be addressed.

 4. The migration assay showed in Figure 4 indicates that the migration increased after treatment with 50ug/ml of AGEs. Why is that? Also, the condition of 200ug/ml seems to have more migration than the condition with 100ug/ml. Are these selected images the most representative?

 5. In Figure 5, the authors show that pretreatment with RAGE agonist prevented the decrease of proliferation and even increased it. In Figure 5A there is shown that 25uM FPS-ZM1 translates to 104% of proliferation. How is this possible? What is the effect of FPS-ZM1 at the baseline proliferation, so cells treated only with BSA?

 6. The same question goes for the survival shown in Figure 6 and migration in Figure 7. What is the effect of the agonist at baseline? Adequate controls should be included.

 7. Is using the AGEs inhibitors proven to prevent or revert the damage caused by ROS in general? Since the authors point towards the importance of ROS in the formation of AGEs in the introduction and discussion, it would be logical to induce ROS by, for instance, hypoxia or H2O2 treatment and show the effect of AGEs inhibitors on CASCs proliferation, survival, and migration.

 8. I think the conclusion should be rephrased. Talking about the therapeutic capacity of CASCs is a little bit far-fudged since it was not studied in this manuscript. As indicated in the title of the manuscript, it is about the properties of the cells that could have potential therapeutic applications.

9. Figures 1, 2, 3, and 4, the error bars are missing from the first bar, and they should be included.

Reviewer 2 Report

Authors investigated to the association of AGEs and CASCs. The results are clear and interesting. However, it needs more detailed experiments. Please answer questions and consider investigating additional experiments.

  1. Introduction
    Please add a summary about the association between AGEs and stem cells. According to author’s introduction, AGEs are just only the products of oxidative stress. It is not clear how AGEs affect stem cell properties.
  2. Materials and Methods
    How can authors identify the proliferation of CASCs?
  3. Results
    1) Please align the brightness of pictures in Figure 4 and 7.
    2) Did authors perform proliferation and survival experiments with over 25uM of FPS-ZM1 (e,g, 50uM)? The proliferation rate is supposed to be increased with over 25 uM of it. Then, it may also affect better prognosis in survival rate.
  4. Discussion
    Since authors cited that “immune cells secrete AGEs”, authors should investigate to illustrate the mechanism as a point of view of immune response. Did authors check chemokines or cytokines? In addition, which kinds of regenerative capacity authors think about CASCs? Please clarify.

Round 2

Reviewer 2 Report

Authors answered well. I would like to accept.